# Investigating the Impacts of UVC Radiation on Natural and Cultured Biofilms: An assessment of Cell Viability

**DOI:** 10.3390/microorganisms11051348

**Published:** 2023-05-20

**Authors:** Cierra R. Braga, Kailey N. Richard, Harrison Gardner, Geoffrey Swain, Kelli Z. Hunsucker

**Affiliations:** Center for Corrosion and Biofouling Control, Florida Institute of Technology, Melbourne, FL 32901, USA

**Keywords:** UVC, biofilms, viability

## Abstract

Biofilms are conglomerates of cells, water, and extracellular polymeric substances which can lead to various functional and financial setbacks. As a result, there has been a drive towards more environmentally friendly antifouling methods, such as the use of ultraviolet C (UVC) radiation. When applying UVC radiation, it is important to understand how frequency, and thus dose, can influence an established biofilm. This study compares the impacts of varying doses of UVC radiation on both a monocultured biofilm consisting of *Navicula incerta* and field-developed biofilms. Both biofilms were exposed to doses of UVC radiation ranging from 1626.2 $\mathrm{mJ}/{\mathrm{cm}}^{2}$ to 9757.2 $\mathrm{mJ}/{\mathrm{cm}}^{2}$ and then treated with a live/dead assay. When exposed to UVC radiation, the *N. incerta* biofilms demonstrated a significant reduction in cell viability compared to the non-exposed samples, but all doses had similar viability results. The field biofilms were highly diverse, containing not only benthic diatoms but also planktonic species which may have led to inconsistencies. Although they are different from each other, these results provide beneficial data. Cultured biofilms provide insight into how diatom cells react to varying doses of UVC radiation, whereas the real-world heterogeneity of field biofilms is useful for determining the dosage needed to effectively prevent a biofilm. Both concepts are important when developing UVC radiation management plans that target established biofilms.

## 1. Introduction

Biofouling is the unwanted recruitment and growth of marine organisms on a submerged substrate. In general, biofouling is a multistep process in which (1) a conditioning film is formed via the accumulation of adsorbed organic molecules on a surface, (2) the primary colonization of bacteria initiates the formation of a biofilm matrix, (3) a secondary colonization of additional microbes such as benthic diatoms occurs (this is referred to as a biofilm or slime layer), and finally, (4) a tertiary colonization occurs, and the beginning of macrofouling (e.g., the recruitment of barnacles, calcareous tubeworms, and encrusting bryozoans) is observed [1,2,3,4]. However, the formation and accumulation of biofilms may determine the subsequent succession of biofouling in the marine environment [4,5]. Biofilms are conglomerates of cells, water, and their secreted extracellular polymeric substances (i.e., substances that enable the biofilm to envelop and anchor itself to the substrate). These secreted substances also result in added protection from environmental stressors such as biocide exposure [1,2,4]. While this matrix is an advantage to the sustenance of biofilms, it can create issues for those trying to minimize the aggregations of biofilms in biofouling management as well as human-health-related concerns (e.g., medical applications) [1,6,7].

Biofilm accumulation on ship hulls is known to increase roughness and viscous drag. In a report from the International Maritime Organization [8], it is stated that a layer of slime as thin as 0.5 mm covering up to 50% of a hull surface could trigger an increase in greenhouse gas emissions in the range of 25% to 30%, depending on the ship’s characteristics, its speed, and other prevailing conditions. Schultz [9] noted that smooth surfaces covered with ≥25% biofilm on average experienced a 65% increase in viscous drag. Hunsucker [10] measured an increase in drag from 0.5 ± 0.04 N to 0.75 ± 0.09 N on a piece of PVC measuring 0.1 m × 0.2 m due to the presence of biofilms. Biofilm accumulation on a DDG frigate can account for an 18% increase in required power when compared to a smooth hull that is free of fouling [9]. In turn, this results in an increase in fuel consumption and greenhouse gas emissions and a reduction in overall efficiency [11,12,13]. The most common technologies used in biofouling management are fouling control coatings containing biocides such as copper and zinc oxide [14,15,16]. With growing concerns for inadvertent impacts on the environment, new fouling management strategies are being developed and implemented that are geared towards being more environmentally friendly [17,18].

The application of ultraviolet light (UVC) radiation (100 nm–280 nm) is presently being investigated as a method of improving biofouling management. UVC radiation is considered the most germicidal portion of the UV spectrum as it has the potential to damage the bonds in DNA, thus causing cells to function improperly, and it can even lead to cell death [13,19]. While the application of UVC radiation does cause physical damage to an organism, it is considered more environmentally friendly when compared to other antifouling systems as it does not create toxic by-products [20]. Typically, UVC radiation has been implemented on instrumentation and sensors in the field for use in marine environments or within the ballast tanks of vessels; however, it has more recently been applied to proof-of-concept studies for more large-scale surfaces such as ship hulls [13,21,22,23]. These studies include the integration of UVC lights into silicone [13]. This LED-embedded silicone technology is unique as it has the potential to be adhered to ship hulls, making it convenient for niche areas such as ballast tanks and sea chests [13]. Other concepts include pairing UVC lamps or LEDs with ROVs for ship hull grooming [22]. Results have indicated that the duo is effective only with biofilms remaining on the surface; however, it was stated that the method is mainly successful for a short term, and it was concluded that longer exposure times may be needed. Usually, these investigations into the applications of UVC radiation are more focused on preventing macrofouling than biofilms; however, as previously mentioned, an established biofilm can lay the foundation for these additional colonizers to accumulate.

Laboratory studies typically differ from field research because it is difficult for field biofilms to thrive in a lab setting due to their different nutrient requirements and optimal environmental parameters [24,25]. The common practice of using monocultures in laboratory settings is widely applied because it is easy to replicate and maintain the resultant biofilms. Unfortunately, it is difficult to compare studies using laboratory biofilm responses to field biofilm applications because there may be multiple response variables in a heterogeneous biofilm that influence the biofilm matrix, including inter- and intra-community member interactions. Meanwhile, cultured biofilms are in a controlled setting and lack diversity. Thus, a two-part study was designed to assess the efficacy of UVC radiation on biofilms naturally accrued in the field and on a mono-specific diatom biofilm cultured in a laboratory. The observations and data acquired through this study can be applied to further investigations in determining the effectiveness of UVC radiation for biofouling management strategies but may also be utilized as an assay for laboratory and field comparisons.

## 2. Materials and Methods

This study compared the effectiveness of applying UVC radiation to monocultured biofilms and heterogeneous field biofilms to determine if the results are comparable to one another. While there was some overlap in the methods, for the ease of reading, the two biofilm methods are separated into cultured and field biofilms.

### 2.1. Cultured Biofilms

#### 2.1.1. Navicula Incerta

*Navicula incerta*, a brackish water species which is found within water temperatures ranging from 18 °C to 20 °C, was used to create a heterogeneous biofilm in a laboratory setting. A starter culture of *N. incerta* was provided by North Dakota State University for Ocean Sciences. Once the starter was received at the Florida Institute of Technology, the cultures were examined to assess the health of the cells prior to splitting them. Under a BSL1 laminar hood (Mo: NU-201-430), 70% ethanol (EtOH) was used to sterilize the hood, bench, and equipment to ensure that there was no contamination to the samples. F/2 medium, purchased from Bigelow Laboratory for Marine Sciences, was used as a nutritional source for *N. incerta*. Once the samples were confirmed to be pure, using an aseptic technique, the *N. incerta* culture was split into four 250 mL flasks to ensure that a sustainable culture would be present for the course of the experiment.

For the splitting process, 75 mL was added to four 250 mL flasks. In addition, 45 mL was added to a transferring tube. The liquid within the starter culture flask was poured out into a waste container. Approximately 10 mL of the F/2 medium within the transferring tube was poured into the starter culture. The sample was swirled around, and the liquid was poured into a waste container. This process was repeated twice. The remaining medium within the transferring tube was poured into the starter culture, swirled around, and was then ready for transfer. Using a sterile cotton swab, *N. incerta* was removed from the bottom of the starter culture flask. A pipette was then used to aspirate the sample approximately 5 times. Once completed, 0.5 mL was then pipetted from the starter culture and placed into each of the 250 mL flasks containing the F/2 medium. Each flask was then covered with foil (to allow air flow but prevent contamination) and placed under a fluorescent lamp to grow (using a cycle of 13 light:11 dark). Once the split was completed, the diatoms were allowed to acclimate and settle out in the beakers. For the experimental procedure, 5 mL of *N. incerta* was pipetted onto microscope slides and allowed to settle out. The microscope slides were held in a 4-well plate. The cultures were allowed to settle for 48 h prior to UVC exposure, which was followed by a series of tests (see below).

#### 2.1.2. UVC Exposure

Five milliliters of *N. incerta* was pipetted onto microscope slides held in a 4-well plate within the BSL1 hood to prevent contamination of the biofilm. An extra 10 µL of F/2 medium was added to the wells to increase the cell density on the microscope slides. *N. incerta* was allowed to settle for 5 days under a fluorescent lamp. The biofilm-covered slides were then taken to a dark room where a PRIME digital lighting timer was used to expose the microscope slides to one of the 5 different UVC frequencies: 0 min (control), 10 min, 20 min, 30 min, 40 min, and 60 min. Based on the duration of exposure per this study, the dose of UVC radiation ranged from 1626.2 $\mathrm{mJ}/{\mathrm{cm}}^{2}$ to 9757.2 $\mathrm{mJ}/{\mathrm{cm}}^{2}$ (Table 1).

The exposure of the samples to UVC radiation was achieved using two 3D printed stands that allowed for the lamp to be placed 25 mm above the microscope slides [27,28,29]. Of the 6 UVC-exposed slides, 4 were used for the 2,3-Bis-(2-Methoxy-4-nitro-5-sulfophenyl)-2H-tetrazolium-5-carboxanilide, disodium salt (XTT) assay (described below). The remaining 2 slides were viewed under a microscope (40×) to determine if there were general community formation and morphological changes due to UVC exposure (e.g., a reduction in size or a loss of organic material).

#### 2.1.3. XTT Assay

Immediately following UVC exposure, an XTT assay (using a Biotium XTT cell viability kit PI-30007, Fremont, CA, USA) was performed in four replicates to determine the concentration of cell death per slide. When the XTT solution is added to viable cells, it is absorbed, giving off an orange coloration [30,31]. Additionally, a higher absorbance depicts a greater presence of viable cells, whereas a low absorbance depicts the presence of fewer metabolically active cells [30,31].

The biofilms were scraped off the microscope slides using a sterile plastic cell lifter and placed in a labeled tube containing 2.4 mL of filtered seawater. A volume of 1.2 mL of XTT reagent was added to each tube and mixed with the biofilm. Blank tubes (control) were filled with 3.6 mL of a mixture of artificial seawater and F/2 medium (culture-grown biofilms only) and 1.2 mL of XTT reagent. Each tube was placed in a dark space for 24 h, allowing for proper absorption by the live cells. Following the 24 h incubation period, the solid material was able to settle out of the samples, and the overlaying supernatant was pipetted out carefully and placed into a plastic disposable cuvette (3.6 mL) to be analyzed in a Hach Dr 6000 spectrophotometer (Loveland, CO, USA) at 492 nm. Control blanks containing just the filtered seawater solution mentioned above were placed inside the spectrophotometer and zeroed out to provide a baseline for the comparison of each sample.

### 2.2. Field Biofilms

Natural biofilms were collected from a static immersion barge located at the Florida Institute of Technology’s Marine Center in Melbourne, FL. This estuarine test site has an average salinity of 16 ± 4.6 ppt and an average water temperature of 27 ± 4.1 °C. Fouling occurs at the test site year-round, with barnacles as the dominant fouling organism. A flow channel system was implemented on the barge to allow for the formation of biofilms in the field without macrofouling coverage overwhelming the test surfaces. The flow channel had a depth of 75 mm, and the average flow rate was 32.7 L/s. The flow channel was able to hold 5 PVC panels (100 mm × 200 mm) that were altered to act as microscope slide holders by attaching rubber strips to the panels’ surfaces. This allowed for six glass microscope slides (25 mm × 75 mm) to be easily inserted and removed for testing (a total of 30 microscope slides). After a week of immersion (January 2021), the slides were removed and exposed to UVC radiation as described below.

Similar to the cultured biofilms, 6 slides containing the field biofilm were exposed in air to UVC light (25-watt Aqua UVC 254 nm lamp) at a distance of 25 mm, using the same setup described previously. A PRIME digital lighting timer was used to expose the microscope slides to one of four doses: 0 min (control), 10 min, 30 min, 45 min, and 60 min for each dose during the course of the study [27,28,29]. Based on the duration of exposure per this study, the dose of UVC radiation ranged from 1626.2 $\mathrm{mJ}/{\mathrm{cm}}^{2}$ to 9757.2 $\mathrm{mJ}/{\mathrm{cm}}^{2}$ (Table 1). Due to the setup of the flow channel, a limited number of microscope slides were able to be used for biofilm formation. Thus, the dose times were selected to favor a range of UVC exposure times. The minimum and maximum UVC exposure times were kept the same to allow for a direct comparison of these values to the cultured biofilms. Following UVC exposure, an XTT assay analysis procedure, as described above in 2.1.3 XTT Assay, was conducted. In addition, two slides (one control slide and one UVC-exposed slide) were randomly selected per exposure time for observation under the microscope to determine if there were general community formation and morphological changes due to UVC exposure (e.g., a reduction in size or loss of organic material). However, the samples were highly congested with various planktonic and benthic diatoms as well as organic matter, which made it difficult to determine if any morphological changes had occurred.

### 2.3. Statistical Analysis

A *t*-test was performed independently on both the cultured and field biofilm data to compare the absorbance readings based on the dose of UVC radiation. Then, a factorial analysis of variance (ANOVA) was carried out to determine if the field and cultured biofilms that were exposed to UVC radiation were significantly different from one another. All analyses were performed using R statistical software version 4.2.3 (2022).

## 3. Results

The samples collected in the field resulted in thicker biofilms when compared to the cultured samples. This may have led to the differences in XTT absorbance between the two methods (Table 2), which the statistical analysis demonstrated was significant (*p* = 0.02). Due to the differences between the two experiments, they will be presented separately below.

### 3.1. Cultured Biofilms

Twenty-four hours after the addition of the XTT solution, the controls were bright orange in color, demonstrating a high level of cell viability. For the controls, the absorbency averaged 0.37 ± 0.15 Au (Table 2). The UVC-exposed *N. incerta* were a faint orange color, and there was no variance among the doses of UVC radiation (*p* > 0.05). The *N. incerta* exposed to 10 min (1626.2 $\mathrm{mJ}/{\mathrm{cm}}^{2}$) and 60 min (9757.2 $\mathrm{mJ}/{\mathrm{cm}}^{2}$) of radiation had the lowest absorbencies of 0.04 ± 0.03 Au and 0.04 ± 0.05 Au, respectively, and the 30-minute exposure (4878.6 $\mathrm{mJ}/{\mathrm{cm}}^{2}$) samples had the highest absorbency of 0.08 ± 0.03 Au (Figure 1). All doses of UVC radiation resulted in significantly less cell viability than the control (*p* < 0.05).

In order to identify morphological changes in the diatom frustule (e.g., reduced size or a loss of organic material), UVC- exposed and non-exposed (controls) cells were viewed under a microscope. The non-exposed *N. incerta* were green in coloration, indicating that they were full of chlorophyll. In addition, all cells were in a valve view (Figure 2). However, this was not the case with the UVC-exposed cells. The *N. incerta* exposed to 60 min (9757.2 $\mathrm{mJ}/{\mathrm{cm}}^{2}$) of UVC radiation were almost clear in color, suggesting that the UVC radiation potentially deteriorated the chlorophyll within the cells. The lower the dose of UVC radiation, the easier it was to identify the chlorophyll within the cells (Figure 2). Furthermore, the majority of cells exposed to UVC radiation reoriented themselves from the valve view to the girdle position. As for the cells that did not reorient themselves, there was no clear reduction in their overall size when compared to the controls. 

### 3.2. Field Biofilms

Two slides (one control slide and one UVC-exposed slide) were viewed under the microscope to identify any morphological changes within the biofilms. When viewing the films, both benthic and planktonic diatoms were encased in the biofilms, creating an obstructed view (Figure 3). For this reason, morphological changes were difficult to identify. Some of the diatom species found within the flow channel included *Stephanopyxis* sp., *Coscinodiscus* sp., *Pleurosigma* sp., *Melosira* sp., *Bacillaria* sp., *Biddulphia* sp., and *Naviculla* sp.

The highest dose of UVC radiation resulted in the lowest absorbance value of 0.42 ± 8.1 × $10^{-4}$ Au (Table 2; Figure 1). The maximum absorbance value seen was at the 30 min (4878.6 $\mathrm{mJ}/{\mathrm{cm}}^{2}$) mark of exposure to UVC radiation was 0.91 ± 5.5 × $10^{-4}$ Au, which was comparable to the control value of 0.89 ± 1.3 × $10^{-3}$ Au (Table 2; Figure 1). Even though there was no significant differences between the control and the dosages, there was an observable trend in which the absorbance (i.e., cell viability) decreased with increasing UVC exposure (*p* = 0.06). However, as stated previously, the congested nature of the diatoms and other biotic matter present on the slides (Figure 3) could have influenced the results, leading to the insignificant values.

## 4. Discussion

To the authors’ knowledge, this is the first UV impact study that compares the cell viability of biofilms cultured in a laboratory setting and field-accrued biofilms. Marine biofilms primarily consist of benthic diatoms [32]; however, when observed closely, the biofilms developed in the flow channel in this study consisted largely of planktonic species. When the microscope slides from the flow channel were evaluated, the field biofilms consisted primarily of large chain-forming planktonic species such as *Stephanopyxis* sp. and *Biddulphia* sp. (~90% of the sample). While there is no objection to planktonic diatoms, benthic diatoms are the primary species found in marine biofilms because they contain a raphe that allows for attachment to surfaces. In contrast, planktonic diatoms are typically chain-forming species that lack the ability to attach to a substrate, making them easy to wash away. For this reason, planktonic diatoms are not considered to be the type of biofilm found under the dynamic flow of a ship’s hull [33]. Previous studies have demonstrated that an increase in water flow over a static community increases species richness, explaining the results from the flow channel seen here [34]. Biofilms form a matrix referred to as EPS (extracellular polysaccharides) that can function as UV protection [2]. This matrix can contribute to communal immunity to UV radiation. In addition, silty mud sediments, similar to the sediments found at the Marine Center field test site, have been found to absorb large amounts of UV radiation [35,36]. The coupling of the biofilm matrix and the presence of sediments could have provided protection to the naturally occurring biofilms, resulting in greater cell viability compared to cultured biofilms. Regarding the cultured biofilms, environmental factors and cell density can be controlled, allowing for clear disruptions to the biofilm due to exposure to UVC radiation to be observed.

Bleaching was observed with the *N. incerta* culture herein when exposed to UVC light, but this was not visible in the field biofilms. Similar results have been identified in the symbiont *Amphistegina gibbose* upon exposure to high intensities (0.0168 W $m^{-2}$) of UVB light, which resulted in the bleaching of 89% of the symbionts [37]. Compared to controls, brown algae, *Kappaphycus alvarezii*, also demonstrated reduced amounts of chlorophyll a when exposed to UVB irradiation [38]. Bleaching was more prominent with higher doses of UVC radiation compared to the lower doses of UVC radiation, which could be attributed to what is called the “package effect”. The package effect is described as the decrease in absorbed pigments (including auxiliary pigments) compared to a cell’s natural absorbance potential [39]. The package effect provides a way for algae to thrive under stress, potentially explaining the slight reduction in color (chlorophyll) when the light exposure was reduced. While bleaching was not noticed in the field-accrued biofilms, it may have been possible that damage to the top layers of the biofilm could have occurred but it was not extensive enough to be seen. Biofilms consist of layers in which the surface film would be subjected to the highest dose of UVC radiation, while the base film is protected and able to thrive [40,41].

Diatoms have the ability to alter their behavior to maximize light retention while avoiding photodamage [42], which could explain the change in the behavior of *N. incerta* observed in this experiment. The *N. incerta* exposed to higher doses of UVC light appeared to reorient themselves. While in the current study, there was no significant difference among the exposure times, both the 10 min (1626.2 $\mathrm{mJ}/{\mathrm{cm}}^{2}$) and 60 min (9757.2 $\mathrm{mJ}/{\mathrm{cm}}^{2}$) exposures during the monoculture portion had the lowest absorbencies observed (0.08 ± 0.03 and 0.04 ± 0.05, respectively). This could possibly indicate that *N. incerta* has a threshold for damage when exposed at 25 mm, which will need to be investigated further. Cohn [43] observed that changes in diatom direction increased with exposure times. Specifically, 30% of cells changed direction when exposed to 10 min (1626.2 $\mathrm{mJ}/{\mathrm{cm}}^{2}$) of irradiance (290 nm), and this increased to 100% after the exposure increased, indicating a threshold similar to *N. incerta*. In both the cultured and field portions, a subtle bell curve was observed when viewing the average absorbances in which the 30 min (4878.6 $\mathrm{mJ}/{\mathrm{cm}}^{2}$) mark was the peak (highest viability). It may be possible that the initial shock of exposure to UVC radiation caused a decline in viability, but the biofilm may have attempted to adjust to its environment, such as through the behavioral changes observed in which the diatoms rotated into the girdle view. This behavioral change may have assisted in allowing for the biofilm to persist; however, after 30 min (4878.6 $\mathrm{mJ}/{\mathrm{cm}}^{2}$) of exposure, the biofilms became overwhelmed by the intensity. This indicates that only the hardier individuals in the biofilm remained by the 60 min point (9757.2 $\mathrm{mJ}/{\mathrm{cm}}^{2}$).

The study performed herein indicated that cell viability was clearly reduced with extended exposures to UVC radiation; however, the absorbance measurements were inconsistent. This can especially be seen when comparing the cultured and field results. The monocultured results demonstrated a significant reduction in cell viability with UVC exposure when compared to the control, but the field results did not demonstrate a significant difference between the control and the cells that received doses of UVC radiation. These inconsistencies from the field results could be the outcome of external factors, such as sediment accumulation, organic material accumulation, community composition variation, etc. When compared to the cultured biofilms, the field-accrued biofilms had relatively high absorbance readings. The addition of sedimentation to the field biofilms during the accrual process is an environmental variable that the cultured biofilms did not experience. The increased amount of sediment on the surface of the microscope slides can act as a shield for the biofilms exposed to UVC radiation, thus allowing for higher cell viability values. Meanwhile, the cultured biofilms were contained within a controlled environment in which external factors such as sediment accumulation did not impact the UVC transmission which, in turn, allowed for more consistent viability readings.

These results provide insight into the relationship between biofilms and UVC exposure for both naturally forming marine biofilms and cultured biofilms. However, the complexity of the accrued field biofilms (e.g., sedimentation, organic material, and multiple community members) made it difficult to understand how cell viability was affected by UVC radiation under these conditions. Meanwhile, the cultured *N. incerta* biofilms indicated that the cell viability was significantly lower than that of the controls, and there may be a threshold for the diatoms’ tolerance. Martínez [44] discovered that select biofilm-forming species are able to recover from UVC exposure as long as the cells are still intact. However, the timeframe of recovery differs by species, which is important to consider when observing naturally occurring field biofilms. This was not measured during the current study and could be of importance for future UVC radiation applications which aim to eliminate established biofilms on ship hulls. Furthermore, when biofilms achieve a certain thickness (5 cm or more), shielding provided by the upper portion of the film reduces the effects of the UVC radiation [45], which could explain why the field-accrued biofilms were less affected by UVC light.

The formation of biofouling communities on a ship hull can spread non-native species, thus potentially altering various ecosystems on a global scale as the ship is underway. Thus, the consequences of accumulated growth have escalated the need for biofouling prevention methods to reduce the negative effects associated with hull fouling. Although biofilms can be considered miniscule at times, they have the potential to cause an increase in fuel consumption which has the potential to add up to approximately USD 1.2M in operating costs per ship per year [46]. Biofilms can also be considered a steppingstone or facilitator for macrofouling organisms, which has been demonstrated for barnacles [47] and tubeworms [48]. Therefore, it is important to understand how biofilms are affected by UVC radiation. Potential changes in community composition via UVC exposure could ultimately cause shifts in the subsequent macrofouling community that may be observed.

There are still several unknowns regarding the effects of exposure to UVC radiation on a biofilm (e.g., exposure rate, dosage, and biofouling composition changes, etc.) which must be considered. These factors could alter the biofouling community present on a surface. Based on this study, it is clear that the use of UVC radiation as an antifouling method for reducing biofilms is effective in both monoculture and field settings. However, only one parameter (cell viability) was observed between field and cultured biofilms. Using additional parameters, such as cell density, chlorophyll production, etc., will enable a better understanding of how biofilms respond to UVC light (both behaviorally and physically). This knowledge will enhance the use of UVC radiation as a preventative measure for biofilm accumulation and the removal of pre-existing biofilms from surfaces, but further research is still needed to confirm these findings as well as to explore the effects of exposure to UVC radiation on other species of diatoms and in different environmental contexts.

## Figures and Tables

**Figure 1 microorganisms-11-01348-f001:**
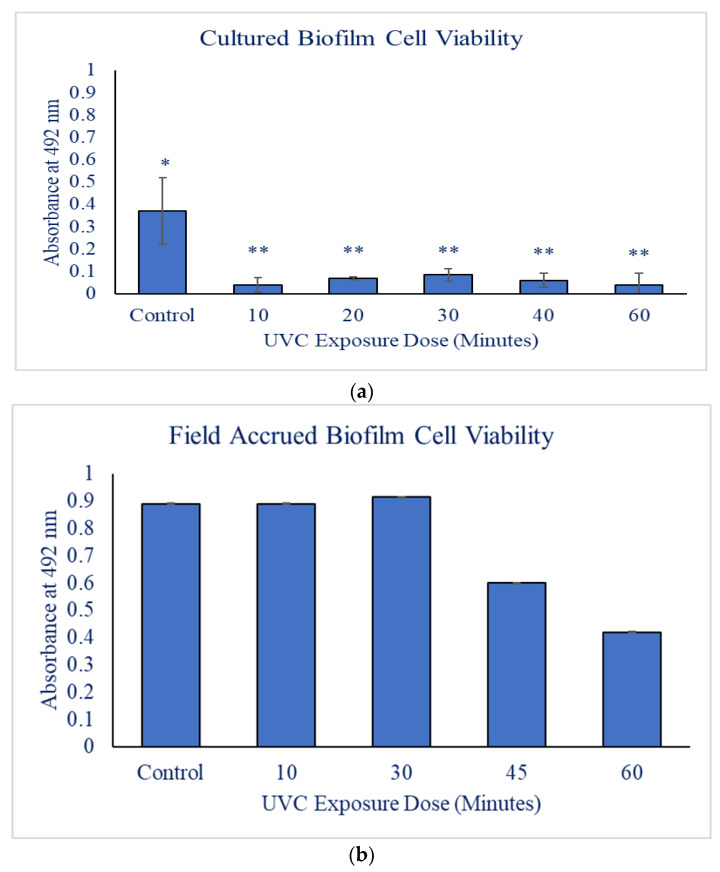
Graph (**a**) is representative of the general trend of declining biofilm viability as UVC exposure increases for the cultured biofilms. Graph (**b**) is representative of the general trend of declining biofilm viability as UVC exposure increases for the field biofilms. * Indicates the significance between the UVC exposure doses tested. ** Indicates that the exposure doses tested were not significantly different from one another.

**Figure 2 microorganisms-11-01348-f002:**
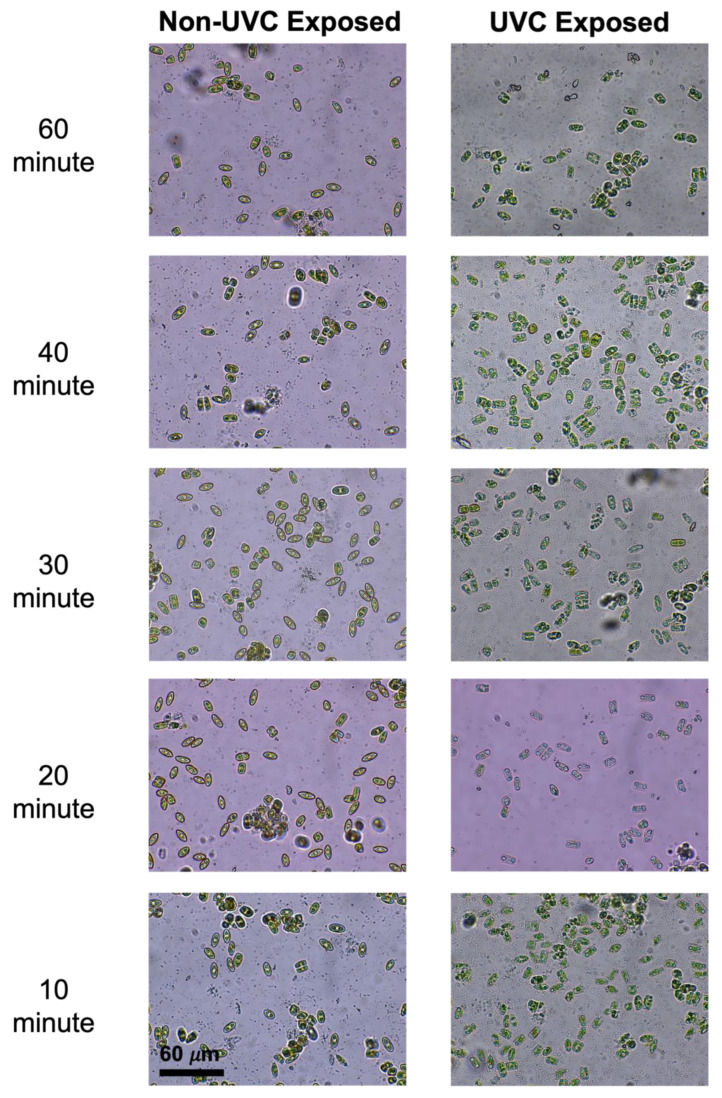
Visual representation of *Navicula incerta* cells under the microscope. The 60 µm scale bar is applied to each of the images. The right depicts the cells not exposed to UVC radiation, and on the left are the UVC-exposed cells after being exposed to one of the five different doses of UVC radiation (60 min, 40 min, 30 min, 20 min, and 10 min).

**Figure 3 microorganisms-11-01348-f003:**
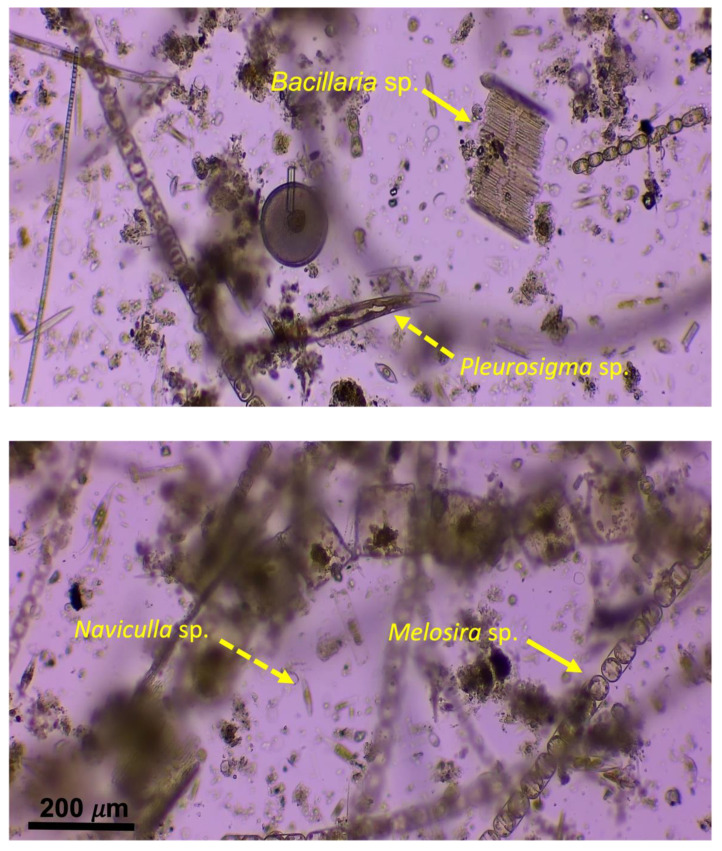
A visual representation of the congested microscope slide from the field. The 200 µm scale bar is applied to each of the images. A sample of the planktonic species observed are indicated by a solid arrow, and benthic species are indicated by a dashed arrow.

**Table 1 microorganisms-11-01348-t001:** This table lists the calculated dose of UVC radiation using the equation D = E × t [26], where D is the dose in $\mathrm{mJ}/{\mathrm{cm}}^{2}$, t is the duration of exposure in seconds, and E is the intensity, measured in $\mathrm{mW}/{\mathrm{cm}}^{2}$.

UV Exposure Time	$\mathrm{Dose}\;(\mathrm{mJ}/{\mathrm{cm}}^{2})$
Control	0
10 min	1626.2
20 min	3252.4
30 min	4878.6
40 min	6504.8
45 min	7317.9
60 min	9757.2

**Table 2 microorganisms-11-01348-t002:** This table lists the average absorbance values recorded during the experiment for field and culture-grown biofilms.

Type	UV Exposure Time	Average XTT Absorbance	Standard Deviation
Field Biofilms	Control	0.89	1.3 × $10^{-3}$
	10 min	0.54	1.1 × $10^{-3}$
	30 min	0.91	5.5 × $10^{-4}$
	45 min	0.60	6.5 × $10^{-4}$
	60 min	0.42	8.1 × $10^{-4}$
Cultured Biofilms	Control	0.37	0.15
	10 min	0.04	0.03
	20 min	0.07	0.01
	30 min	0.08	0.03
	40 min	0.06	0.03
	60 min	0.04	0.05

## Data Availability

Requests for data may be made to the corresponding author of this article.

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
