# Peer review of "Investigating the Impacts of UVC Radiation on Natural and Cultured Biofilms: An assessment of Cell Viability"

_microorganisms, 2023, doi:10.3390/microorganisms11051348_

Round 1
Reviewer 1 Report
This manuscript brings interesting results on the effects of UVC radiation exposure on marine biofilms, specifically focusing on the behavior and viability of the diatom N. incerta. The article provides insights into how UVC radiation affects diatoms in both monoculture (cultured) and field (natural) conditions. One of the strengths of the article is that it presents experimental data on the response of N. incerta to UVC radiation exposure, including absorbance measurements and observations of diatom behavior. The article also compares the results obtained from monoculture experiments with field observations, providing a broader context for the findings. However, there are also some limitations that, once addressed, will improve the current version of the manuscript. See below my comments and suggestions:
Introduction
The Introduction provides a clear and concise overview of the topic of biofouling and its impact on marine environments. The authors have effectively highlighted the different stages of biofouling and the role of biofilms in the formation of marine organisms on submerged substrates. In addition, several examples of how biofouling can negatively impact the efficiency of ships, resulting in increased fuel consumption and greenhouse gas emissions.
The discussion on the use of ultraviolet light (UVC) radiation as a potential solution for biofouling management is interesting and timely. However, the authors could have provided more details on the current state of research on this topic. For instance, they mention that UVC has been implemented in the field for instrumentation and sensors, but more recent studies have applied it to proof-of-concept studies for larger surfaces like ship hulls. It would be helpful to provide some details on these proof-of-concept studies, such as the specific methods used and the effectiveness of UVC in preventing biofouling on ship hulls.
In addition, the authors have mentioned that laboratory studies differ from field research, but it is unclear how this impacts the validity of laboratory studies in the context of biofouling research. It would be helpful to expand on this point and discuss the limitations of laboratory studies in understanding biofouling in natural environments.
Methods
The methodological paragraphs provide a detailed description of the process used to create and treat the Navicula incerta biofilms in the laboratory. The use of a starter culture from North Dakota State University and examination of the health of the cells prior to splitting them are good practices to ensure consistency and reliability of the results.
The addition of F2 medium to increase cell density on the microscope slides is a useful detail, but it is not clear what the concentration of the medium was. Additionally, it would be helpful to know the specific manufacturer and composition of the F2 medium used in the experiment. In addition, the paragraph could benefit from more details on how the biofilm-covered slides were handled during and after the exposure to UVC. For example, it would be useful to know if the slides were washed or rinsed, or if they were observed immediately after exposure.
2.1.3. XTT Assay
The use of four replicates and control blanks is appropriate to ensure the accuracy and validity of the results. The methods used for scraping the biofilms, adding the XTT reagent, and incubation are also clearly described. However, some important details are missing that could be useful for other researchers to replicate this experiment, such as the specific brand or source of XTT reagent used, the type of cuvette used for spectrophotometric analysis, and the criteria used for determining the cutoff value for the absorbance.
2.2. Field Biofilms
The description of the test site and the flow channel system used for biofilm formation is clear and informative. However, there are a few areas that could be improved.
Firstly, it would be helpful to provide more detail about the composition of the biofilms collected from the field. Specifically, it would be useful to know what organisms were present in the biofilms, in addition to the dominant fouling organism, barnacles. This information could help readers understand how representative the collected biofilms are of natural biofilms in the area.
Secondly, it would be beneficial to provide more information about the observation of the biofilms under the microscope. Specifically, it would be helpful to know what magnification was used, and whether any morphological changes were observed in the biofilms because of UVC exposure. Additionally, it would be useful to know whether any other observations were made (e.g. changes in community structure).
Discussion
The discussion provides an in-depth analysis of the experiment conducted on marine biofilms and their response to UVC radiation. However, there are some parts that may be improved, as detailed below:
One major limitation is the inconsistency in the results obtained from the cultured (monoculture) experiments and the field observations. The authors acknowledge this inconsistency but do not provide a clear explanation for it, which raises questions about the validity and reliability of the findings and suggests that further investigation may be needed to understand the differences between cultured and field biofilms in response to UVC exposure.
The discussion lacks enough evidence to support some of the claims made. For instance, the claim that "biofilms developed in the flow channel in this study consisted largely of planktonic species" needs more evidence to be convincing.
Although the study's results are mentioned, there is a need for a more detailed comparison with the results of other studies to establish the significance of the findings.
The discussion needs a more detailed interpretation of the results. For example, the authors mention that the diatoms may have reoriented themselves in response to UVC radiation, but it does not provide a clear explanation of why or how this happened.
While the study briefly mentions some of the potential implications of the findings (e.g., the threshold for damage in diatoms, and the role of biofilm behavior in response to UVC exposure), it does not provide a comprehensive discussion of the significance and implications of the results. Further elaboration on the ecological and environmental implications of the findings would enhance the overall impact of the article.
Please add more specific and concise conclusions that summarize the key findings of the study. The discussion could also provide recommendations for future research in the field, e.g. “further research is needed to confirm these findings and explore the effects of UVC exposure on other species of diatoms and in different environmental contexts.”
Author Response
The authors would first like to thank the reviewer for taking the time to review the manuscript and provide their constructive feedback.
In response to:
For instance, they mention that UVC has been implemented in the field for instrumentation and sensors, but more recent studies have applied it to proof-of-concept studies for larger surfaces like ship hulls. It would be helpful to provide some details on these proof-of-concept studies, such as the specific methods used and the effectiveness of UVC in preventing biofouling on ship hulls.
- Added L64-70: “These studies include the integration of UVC lights into silicon [13]. This LED embedded silicone application is unique as it has the potential to be adhered to ship hulls making it convenient for niche areas such as ballet tanks and sea chest [13]. Other concepts include pairing UVC lamps or LEDs with ROVs for ship hull grooming [22]. Results have indicated that the duo is effective with only biofilms remaining on the surface; however, it was stated that the method is mainly successful for a short term and it was concluded that longer exposure times may be needed.”
In addition, the authors have mentioned that laboratory studies differ from field research, but it is unclear how this impacts the validity of laboratory studies in the context of biofouling research. It would be helpful to expand on this point and discuss the limitations of laboratory studies in understanding biofouling in natural environments.
- Added to L74: “Unfortunately, it is difficult to compare studies using laboratory biofilm responses to field biofilm applications because in a heterogeneous biofilm there may be multiple response variables influencing the biofilm matrix including inter- and intra-community member interactions; meanwhile, cultured biofilms are in a controlled setting and lack diversity.”
Methods
The addition of F2 medium to increase cell density on the microscope slides is a useful detail, but it is not clear what the concentration of the medium was. Additionally, it would be helpful to know the specific manufacturer and composition of the F2 medium used in the experiment. In addition, the paragraph could benefit from more details on how the biofilm-covered slides were handled during and after the exposure to UVC. For example, it would be useful to know if the slides were washed or rinsed, or if they were observed immediately after exposure.
- L116: “10 uL” of F2 medium was added
- L133: “Immediately” following UVC exposure
- L139: Biofilms were scrapped off using “a plastic cell lifter”
2.1.3. XTT Assay
The use of four replicates and control blanks is appropriate to ensure the accuracy and validity of the results. The methods used for scraping the biofilms, adding the XTT reagent, and incubation are also clearly described. However, some important details are missing that could be useful for other researchers to replicate this experiment, such as the specific brand or source of XTT reagent used, the type of cuvette used for spectrophotometric analysis, and the criteria used for determining the cutoff value for the absorbance.
- Added the source of the XTT reagent: “XTT assay (Biotium XTT cell viability kit PI-30007)”
- Added information on the cuvettes used: “a plastic disposable cuvette (3.6 mL)”
- Expanded on splitting process: “For the splitting process, 75 mL was added to four 250 mL flasks. In addition, 45 mL was added to a transferring tube. The liquid within the starter culture flask was poured out into a waste container. Approximately 10 mL of the F/2 medium within the transfer-ring tube was poured in the starter culture. The sample was swirled around, and the liquid was poured into a waste container. This process was repeated twice. The remaining medium within the transferring tube was poured into the starter culture, swirled around and was ready for transferring. Using a sterile cotton swab, N. incerta was removed from the bottom of the starter culture flask. A pipette was then used to aspirate the sample approximately 5 times. Once completed, 0.5 mL was then pipetted from the starter culture and placed into each of the 250 mL flask containing the F/2 medium. Each flask was then covered with foil (to allow air flow but prevent contamination) and placed under a fluorescent lamp to grow (13 light:11 dark cycle). Once the split was completed the diatoms were allowed to acclimate and settle out in the beakers. For the experimental procedure, 5 mL of N. incerta was pipetted onto microscope slides and allowed to settle out. Microscope slides were held in a 4-well plate. Cultures were allowed to settle for 48 hours prior to UV-C exposure and followed by a series of testing (see below).”
2.2. Field Biofilms
Firstly, it would be helpful to provide more detail about the composition of the biofilms collected from the field. Specifically, it would be useful to know what organisms were present in the biofilms, in addition to the dominant fouling organism, barnacles. This information could help readers understand how representative the collected biofilms are of natural biofilms in the area.
- Added “Some of the diatom species found within the flow channel include Stephanopyxis, Coscinodiscus sp., Pleurosigma sp., Melosira sp., Bacillaria sp., Biddulphia sp., and Naviculla sp.
Secondly, it would be beneficial to provide more information about the observation of the biofilms under the microscope. Specifically, it would be helpful to know what magnification was used, and whether any morphological changes were observed in the biofilms because of UVC exposure. Additionally, it would be useful to know whether any other observations were made (e.g. changes in community structure).
- L229: States “morphological changes were difficult to identify”
- L129-130: Added in magnification – “slides were viewed under a microscope (40x) to determine if there were general community formation and morphological changes due to UVC exposure”
Discussion
One major limitation is the inconsistency in the results obtained from the cultured (monoculture) experiments and the field observations. The authors acknowledge this inconsistency but do not provide a clear explanation for it, which raises questions about the validity and reliability of the findings and suggests that further investigation may be needed to understand the differences between cultured and field biofilms in response to UVC exposure.
- To expand on the reasons for inconsistencies between cultured and field biofilms, the authors added: “The addition of sedimentation to the field biofilms during the accrual process is an environmental variable that the cultured biofilms did not experience. The increased amount of sediment on the surface of the microscope slides can act as a shield for the biofilms exposed to UVC thus allowing for higher cell viability values. Meanwhile, the cultured biofilms were contained within a controlled environment where external factors like sediment accumulation did not impact the UVC transmission which in turn allowed for more consistent viability readings.”
The discussion lacks enough evidence to support some of the claims made. For instance, the claim that "biofilms developed in the flow channel in this study consisted largely of planktonic species" needs more evidence to be convincing.
- Added species information to L229-231 to enhance the claim for more planktonic species observed: “Some of the diatom species found within the flow channel include Stephanopyxis, Coscinodiscus sp., Pleurosigma sp., Melosira sp., Bacillaria sp., Biddulphia sp., and Naviculla sp.
- Added in Discussion: “When the microscope slides from the flow channel were evaluated, the field biofilms consisted primarily of large chain forming planktonic species such as Stephanopyxis and Biddulphia sp. (~90% of the sample). While there is no objection to planktonic diatoms, benthic diatoms are the primary species found in marine biofilms because they contain a raphe that allows for attachment to the surface. In contrast planktonic diatoms are typically chain forming species that lack the ability to attach to a substrate making them easily washed away. For this reason, planktonic diatoms are not considered for the type of biofilm experienced under dynamic flow of a ship hull [33].
Although the study's results are mentioned, there is a need for a more detailed comparison with the results of other studies to establish the significance of the findings.
- Added to the discussion: “To the authors’ knowledge, this is the first UV impact study that compares the cell viability of biofilms cultured in laboratory setting and field accrued biofilms.”
The discussion needs a more detailed interpretation of the results. For example, the authors mention that the diatoms may have reoriented themselves in response to UVC radiation, but it does not provide a clear explanation of why or how this happened.
- The source [42] was to provide comparative information on the diatom reorientation: “Diatoms have the ability to alter their behavior to maximize light retention while avoiding photodamage [42], which could explain the change in behavior of incerta observed in this experiment.”
While the study briefly mentions some of the potential implications of the findings (e.g., the threshold for damage in diatoms, and the role of biofilm behavior in response to UVC exposure), it does not provide a comprehensive discussion of the significance and implications of the results. Further elaboration on the ecological and environmental implications of the findings would enhance the overall impact of the article.
- Added “The formation of biofouling communities on a ship hull can spread non-native species thus potentially altering various ecosystems on a global scale as a ship is underway. Thus, the consequences of accumulated growth have escalated the need for biofouling prevention methods to reduce the negative effects associated with hull fouling. Although biofilms can at times be considered miniscule, they have the potential to cause an increase in fuel consumption which has the potential to add up to approximately $1.2M in operating costs per ship per year [46]. Biofilms can also be considered as a steppingstone, or facilitator, for macrofouling organisms which has been demonstrated for barnacles [47] and tubeworms [48]. Therefore, it is important to understand how biofilms are affected by UVC. Potential changes in community composition via UVC exposure could ultimately cause shifts in the subsequent macrofouling community may be observed.”
- Added “While there are still several unknowns about the effects of UVC exposure on a biofilm (g. exposure rate, dosage, biofouling composition changes, etc.) which still need to be considered. These factors could alter the biofouling community present on a surface.”
Please add more specific and concise conclusions that summarize the key findings of the study. The discussion could also provide recommendations for future research in the field, e.g. “further research is needed to confirm these findings and explore the effects of UVC exposure on other species of diatoms and in different environmental contexts.”
- Added “This knowledge will enhance the use of UVC as a preventative measure for biofilm accumulation and the removal of pre-existing biofilms from surfaces, but further research is still needed to confirm these findings as well as explore the effects of UVC exposure on other species of diatoms and in different environmental contexts.”
Reviewer 2 Report
The article is devoted to the influence of ultraviolet C on biofilms. In my opinion, the work obtained very useful data. The data obtained are consistent with the methods used. It should be noted one hundred article is well written, as well as the quality of the graphic material. In my opinion, the article can be published as presented.
Author Response
The authors are thrilled that you enjoyed the manuscript and thought that it did not need to be modified before publication. Thank you for taking the time to read through it and provide your thoughts. The authors are very grateful.
Reviewer 3 Report
The manuscript “Investigating UVC Impacts on Natural and Cultured Biofilms: An assessment on cell viability”, is a preliminary study, that describes the effects of UVC radiation on the biofilm both in vivo and in vitro by analyzing a single parameter - cell viability, tested by XTT assay. The introduction is well written, presenting the importance and the current state of knowledge in this field, but in the other sections there are some issues that need to be addressed during the revision and are listed below:
row 86 - the time (ex., no of days/ no of hours) in which the biofilm was created in the laboratory must be written
row 92 – aseptic technique – this should be described, or a bibliography resource should be given.
rows 199, 206 – please indicate by barr scale
row 206 – indicate with arrows the benthic and planktonic diatoms.
The results are well presented and justified, and the conclusions support the continuation of the study by measuring other parameters.
Author Response
The authors would first like to thank the reviewer for taking the time to review the manuscript and provide their constructive feedback.
In response to:
row 86 - the time (ex., no of days/ no of hours) in which the biofilm was created in the laboratory must be written
- Lines 100 – 102: describes that “incerta was allowed to settle for 5 days under a fluorescent lamp”
row 92 – aseptic technique – this should be described, or a bibliography resource should be given.
- Lines 91-94: Added “Under a BSL1 laminar hood (Mo: NU-201-430), 70% ethanol (EtOH) was used to sterilize the hood, bench and equipment to ensure there was no contamination to the samples. F/2 medium, purchased from Bigelow Laboratory for Marine Sciences, was used as a nutritional source for incerta.”
rows 199, 206 – please indicate by barr scale
- Added barr scale to Figures 2 and 3.
row 206 – indicate with arrows the benthic and planktonic diatoms.
- In Figure 3, arrows were added to point out some of the benthic and planktonic diatoms seen in the sample. Added into figure caption “A sample of the planktonic species observed are indicated by a solid arrow and benthic species are indicated by a dashed arrow.”
Reviewer 4 Report
In this article, the authors investigate the effect of UVC on natural and cultured biofilms in terms of cell viability. The authors studied the effect of UVC on field biofilms and monocultured biofilms. The data from this study would be useful in developing strategies based on UVC to target biofilms. However, the data in this study is very preliminary and further studies would be highly desired considering other parameters apart from cell viability.
Minor comments:
L107: XTT à full form
L173, 174: N. incerta (italics). Please check and correct it throughout the manuscript.
Since table 2 and Fig. 1 contain the same data, table 2 could be moved as supplementary information.
Figure legends: Scientific names in italics
Author Response
The authors would first like to thank the reviewer for taking the time to review the manuscript and provide their constructive feedback.
In response to:
L107: XTT à full form
- Line 112: Added “2,3-Bis-(2-Methoxy-4-nitro-5-sulfophenyl)-2H-tetrazolium-5-carboxanilide, disodium salt (XTT)”
L173, 174: N. incerta (italics). Please check and correct it throughout the manuscript.
- The authors have gone through the manuscript and italicized all Navicula incerta present that were not italicized at the time.
Since table 2 and Fig. 1 contain the same data, table 2 could be moved as supplementary information.
- The authors appreciate the reviewer’s feedback and thoughts on moving Table 2 to supplementary information. However, the authors would like to keep Table 2 in the text along with Figure 1 to emphasize the differences in values demonstrated if acceptable.
Figure legends: Scientific names in italics
- The scientific name, Navicula incerta, in the caption/legend of Figure 2 has been italicized.
Round 2
Reviewer 3 Report
I recommend this work for publication in this revised form!
Reviewer 4 Report
L253: To the authors' knowledge --> You mean readers' knowledge?